# Assessment of the Interferon-Lambda-3 Polymorphism in the Antibody Response to COVID-19 in Older Adults Seropositive for CMV

**DOI:** 10.3390/vaccines11020480

**Published:** 2023-02-18

**Authors:** Ariane Nardy, Camila Tussato Soares Camargo, Yasmim Faustina Castro de Oliveira, Fernanda Cristina da Silva, Millena Soares de Almeida, Fernanda Rodrigues Monteiro, Brenda Rodrigues Silva, Jônatas Bussador do Amaral, Danielle Bruna Leal Oliveira, Edison Luiz Durigon, Guilherme Pereira Scagion, Vanessa Nascimento Chalup, Érika Donizetti Candido, Andressa Simões Aguiar, Neil Ferreira Novo, Marina Tiemi Shio, Carolina Nunes França, Luiz Henrique da Silva Nali, André Luis Lacerda Bachi

**Affiliations:** 1Faculty of Pharmacy, Campus, Santo Amaro University, São Paulo 04829-300, Brazil; 2Post-Graduation Program in Health Sciences, Santo Amaro University (UNISA), São Paulo 04829-300, Brazil; 3ENT Research Laboratory, Department of Otorhinolaryngology—Head and Neck Surgery, Federal University of Sao Paulo (UNIFESP), São Paulo 04021-001, Brazil; 4Hospital Israelita Albert Einstein, São Paulo 05652-900, Brazil; 5Laboratory of Clinical and Molecular Virology, Department of Microbiology, Institute of Biomedical Science, University of São Paulo, São Paulo 05508-060, Brazil; 6Scientific Platform Pasteur, University of São Paulo, São Paulo 05508-060, Brazil; 7Infection Control Service, São Luiz Gonzaga Hospital of Santa Casa de Misericordia of São Paulo, São Paulo 02276-140, Brazil

**Keywords:** cytomegalovirus, SARS-CoV-2, immunoglobulins, neutralizing antibody, outbreak, allele

## Abstract

Background: Here, we investigated the impact of IFN-lambda-3 polymorphism on specific IgG responses for COVID-19 in older adults seropositive for CMV. Methods: Blood samples of 25 older adults of both sexes were obtained at three different times: during a micro-outbreak (MO) of SARS-CoV-2 in 2020; eight months after (CURE); and 30 days after the administration of the second dose of ChadOx-1 vaccine (VAC). The specific IgG for both SARS-CoV-2 and CMV antigens, neutralizing antibodies against SARS-CoV-2, and also the polymorphism profile for IFN-lambda-3 (rs12979860 C > T) were assessed. Results: Higher levels of specific IgG for SARS-CoV-2 antigens were found in the MO and VAC than in the CURE time-point. Volunteers with specific neutralizing antibodies against SARS-CoV-2 showed better specific IgG responses for SARS-CoV-2 and lower specific IgG levels for CMV than volunteers without specific neutralizing antibodies. Significant negative correlations between the specific IgG levels for SARS-CoV-2 and CMV were found at the MO time-point, as well as in the group of individuals homozygous for allele 1 (C/C) in the MO time-point and heterozygotes (C/T) in the CURE time-point. Conclusion: Our results suggested that both CMV seropositivity and the homozygosis for allele 1 (C/C) in IFN-lambda-3 gene can negatively impact the antibody response to COVID-19 infection and vaccination in older adults.

## 1. Introduction

Since the World Health Organization (WHO) declared on March 11, 2020, that the world was facing a new pandemic caused by SARS-CoV-2, which causes the “coronavirus disease-2019” or COVID-19 [1,2]. Systematic reviews showed that the main population affected by this infection were individuals more than 50 years old, preferentially men, and that most of the deaths were associated with individuals over 60 years old [3,4,5].

In this respect, it has been postulated that one of the crucial factors that leads the older adult’s population to present increased infection rate, severity, and lethality regarding SARS-CoV-2 is associated with the occurrence of immunosenescence, a phenomenon characterized by the decline in immune function associated with aging, which, among other aspects, increases susceptibility to infections [6,7,8].

In agreement with the literature, during aging, both T and B cells’ repertoire in the periphery are altered to be restricted to memory cells in association with a reduced number of naive cells [9]. Together, these aspects can substantially impair the response against a new immunological challenge, such as SARS-CoV-2 and/or vaccination [10]. Particularly in terms of B cell activity, it was reported that neutralizing antibody (IgM and IgG) responses after influenza-virus vaccination in older adults showed a significant decrease not only in terms of their lower production but also in terms of their shorter half-life and low affinity for antigens [11]. Furthermore, it is of utmost importance to mention that the influenza virus vaccine-induced immunogenicity in older adults is reduced (only 30–40%) [12]. In fact, when the influenza vaccine’s composition coincides with the circulating strain of this virus, the effectiveness of vaccination in healthy adults is between 70% and 90%, but falls to between 30% and 50% in people >60 years old [13]. In addition, it was also reported that seasonal influenza vaccine effectiveness in older adults (>65 years old) was at about 49%, whereas for healthy younger/adults (18–64 years old) it was closer to 59% [14]. In terms of COVID-19, currently, several studies have focused on the assessment of the immunogenicity of the different vaccines for COVID-19, and, in a general way, there are lower responses in older adults as compared to younger adults’ results, regardless of the vaccine evaluated [15,16].

Regarding immunosenescence development, it has been postulated that the occurrence of another phenomenon associated with aging called “inflammaging”, which, in accordance with Franceschi et al. [17], can be described as a chronic, systemic, sterile, low-grade inflammation related to aging, could be a corollary factor to promote the decrease in immune responses in older adults [18].

Among some situations that can favor the development of inflammaging, it is proposed that the reactivation of cytomegalovirus (CMV) infection, a member of the Herpesviridae family, can be involved [19]. In this sense, it is paramount to highlight that CMV is exclusively associated with its host through the immune/inflammatory system, since this virus not only utilizes myeloid cells as its main reservoir, but also requires the host’s inflammatory response to perpetuate its life cycle and prevent its elimination [20]. Hence, pro-inflammatory cytokines can act by reactivating the virus from its latency state to a situation of active viral replication, and repeated cycles of asymptomatic reactivation and replication of CMV amplify the systemic pro-inflammatory state of individuals [17,21], especially in the older adult population Thus, it is considered a component of the immunological risk phenotype in older adults [22,23]. 

Concerning the relationship between respiratory diseases and CMV infection, a study by Johnstone et al. [24] showed that older adults living in nursing homes with a high number of CMV-reactive CD4+ T cells presented a higher risk of developing viral respiratory disease. Furthermore, the same authors demonstrated that, among the 1072 older adults who participated in this study, the positivity for respiratory viruses was: influenza (24%), RSV (14%), seasonal coronavirus (OC43, NL63, 229E and HKU1) (32%), rhinovirus (17%), human metapneumovirus (9%), and parainfluenza (5%). Based on these data, in this group of older adult individuals with a high number of CMV-reactive CD4+ T cells, the most prevalent respiratory virus was seasonal coronavirus. 

Control or maintenance of viral infections by the host depends on the presence of certain cytokines, especially those related to the Th1 profile, especially in the interferon (IFN) family [25]. In fact, type I (known as alpha and beta), type II (known as gamma), and type III (known as lambda) IFNs are essential for establishing a robust viral resistance and activating adaptive immune responses against viruses [26]. 

Specifically, in terms of IFN-lambda, it has been reported that this cytokine, and also the single nucleotide polymorphisms (SNPs) linked to the cytokine IFNλ3 (also known as IL28B), can be involved not only in the spontaneous clearance of Hepatitis C virus (HCV) infections, but also in the higher rates of sustained virological responses associated with combined antiviral therapy [27,28,29]. In an interesting way, patients infected with HCV who carried the CC allele at the IFNλ3 polymorphic site presented a higher probability (3 times more) to eradicate this infection (both spontaneously and after antiviral therapy) than the individuals who showed the T genotype [28,29]. 

Particularly in SARS-CoV-2 and CMV infections, an interesting point related to IFN action deserves to be cited; whereas the treatment with IFN-lambda (type III IFN) was able to significantly reduce SARS-CoV-2 replication both in vitro, using colon-derived cell lines and primary colon organoids [30], as well as in vivo, using experimental models with mice [31], this same IFN-type presents a significant action in the control of CMV infection since, as mentioned by Sezgin et al. [32], the polymorphisms in the IFN-lambda 3/4 region can be involved not only in the control of but also in the susceptibility to CMV clearance.

Since the pandemic began, it has been suggested that CMV infection, as well as its seropositivity, could interfere with the immune response to SARS-CoV-2 infection in the older adult population [33,34]. However, until now, there have been few studies that present results about this issue. Furthermore, the association with type III IFN in this context is still unclear. Therefore, based on these pieces of information, we aimed to investigate the impact of polymorphism for interferon-lambda on SARS-CoV-2-specific antibody responses in an older adult group not only infected with but also vaccinated against COVID-19 and who were seropositive for CMV.

## 2. Materials and Methods

### 2.1. Study Design

In order to perform the present study, 25 volunteers (aged between 51 and 85 years old) of both sexes (19 men and 6 women) were enrolled. All were residents of the “HOSPITAL GERIÁTRICO E DE CONVALESCENTES DOM PEDRO II”, belonging to the Health Department of the São Paulo State, Brazil. It is noteworthy to mention that only two volunteers presented with ages below 60 years old. All the participants were infected with SARS-CoV-2 in March-April 2020 during a micro-outbreak in this hospital, and all were immunized for COVID-19 with the ChadOx-1 (AstraZeneca/Oxford) vaccine one year after in 2021.

The main symptoms associated with SARS-CoV-2 infection were obtained, and as reported, all the participants presented mild to moderate symptoms. It is paramount to clarify that the symptoms classification followed the “Guidelines for the Management of Patients with COVID-19” provided by the Brazilian Ministry of Health [6], which are in agreement with the World Health Organization [7] for COVID-19.

None of the volunteers were infected with HIV, or presented any neurological diseases or cancers, and none were submitted to convalescent plasma and/or corticosteroid therapy or were under any other anti-inflammatory medications at some stage of the study. All the volunteers signed the consent form previously approved by the Ethics and Research Committee from the University of São Paulo (USP, under number 36011220.0000.0081) and from Santo Amaro University (UNISA, under number 4.350.476). It is worth mentioning that the study was performed in agreement with the Declaration of Helsinki.

### 2.2. SARS-CoV-2 Diagnosis

Naso-oropharyngeal swab samples from 25 volunteers were used to perform the SARS-CoV-2 diagnosis through the real-time PCR (RT-PCR) test, according to Corman et al. [35] after protocol modification as follows. The total RNAs contained in the naso-oropharyngeal samples were extracted on the MagMAX™ Express (Applied Biosystems, Foster City, CA, USA) with Kit MagMAX™ Viral/Pathogen II (MPV II) and used in the RT-PCR test performed on an ABI 7300 machine and the AgPath-ID One-Step RT-PCR master mix kit (Applied Biosystems Inc., Wakefield, RI, USA). Briefly, the materials to extract genetic material used were: 130 µL of Elution Solution (MPV II), 500 µL de Wash Solution (MPV II), 1000 µL of 80% ethanol, 200 µL sample with 300 µL of Binding Solution plus 20 µL of magnetic Binding Beads (MPV II), and 10 µL de Proteinase K (MPV II). Subsequently, regarding SARS-CoV-2 diagnoses across rt-PCR, each reaction contained: 5 μL of extracted material diluted in a 10 μL mix containing 3.5 μL of OneStep Buffer ([5×], [12.5 mM] of MgCl_2_, Tris Cl, KCl, (NH_4_)_2_SO_4_), 5.5 μL Nuclease-Free Water (UltraPureTM DEPC), [10 pM] of each primer (E-Sarbeco F1 and R2), [7.5 pM] of probe (E-Sarbeco P1) target with FAM, and 0.3 μL of AgPath Enzyme totalizing 15 μL final volume. The detection of nucleic acid targets was accomplished following thermal cycling conditions: one cycle at 50 °C for 20 min and another at 95 °C for 10 min, followed by 45 cycles at 95 °C for 10 s, 58 °C for 10 s, and 72 °C for 33 s. As positive controls, we used a clinical isolated in Vero-E6 cell culture (SARS-CoV-2/SP02/human2020/Br, GenBank accession number MT126808.1), and water as a negative control.

### 2.3. Blood Samples Collection

Blood samples were collected on three different occasions: during a micro-outbreak of COVID-19 in the hospital (MO), eight months after this micro-outbreak (CURE), and thirty days after the administration of the second dose of ChadOx-1 vaccine (VAC). Concerning the blood sampling during the MO, it is noteworthy to mention that it was carried out from 1 to 2 days after the onset of the symptoms. The samples were collected in tubes containing anticoagulant EDTA, and the plasma obtained after the tubes´ centrifugation (2000 rpm, 10 min, at 4 °C) was aliquoted and stored at −80 °C for further analysis of specific IgG for SARS-CoV-2 and CMV.

After the plasma separation, the blood samples were mixed 1:1 with a phosphate-saline buffer (PBS 1X, pH = 7.3), and submitted to the isolation of the peripheral blood mononuclear cells (PBMCs) using Ficoll-Hypaque (GE Healthcare Bio-Sciences AB, Uppsala, Sweden). A quantity of 5 × 10^5^ PBMCs were separated and stored in a freezing medium at −80 °C for the further polymorphism of IFN-lambda assessment.

### 2.4. COVID-19 Vaccination

All volunteers were submitted for vaccination against COVID-19 in 2021. They received two doses of the ChadOx-1 vaccine that was available through “Fundação Oswaldo Cruz (FIOCRUZ)”, Brazil. Blood samples were collected 30 days after they had received the second dose of the vaccine.

### 2.5. Determination of Specific IgG for SARS-CoV-2

Specific IgG for SARS-COV-2 was detected using an ELISA test, following the described process, in our group [36,37]. Briefly, 96-well plates (Corning, New York, NY, USA) were coated with an equimolar mixture of antigens (0.12 μg/mL in sodium carbonate–sodium bicarbonate buffer, including both nucleoprotein (N) and spike (S) antigens) and incubated overnight. After this time, unspecific binding of antibodies was avoided by blocking it with the buffer PBS-BSA-T containing 1% of bovine fetal serum (Invitrogen by Thermo Fisher Scientific, Vienna, Austria) in PBS (1X, pH: 7.3) + 0.05% of Tween 20 (Synth, Diadema, Brazil) at 37 °C for 2 h. After the washing step with a PBS-T solution (PBS 1X, pH: 7.3 + 0.05% of Tween), 100 μL of plasma (diluted at 1:8000) was added and incubated for 1 h at 37 °C. After a new round of washing, the secondary antibody (goat anti-human IgG, Sigma-Aldrich Co., Deisenhofen, Germany) conjugated with horseradish peroxidase diluted at 1:10,000 was added and incubated for 1 h at 37 °C. After the washing step, 100 μL of TMB solution (3.3′.5.5′- tetramethylbenzidine. Thermo Scientific, MA, USA) was added and incubated for 10 min at room temperature avoiding direct exposure to light. The reaction was stopped by adding a solution of sulfuric acid (0.2 N) to each well, and the optical density at 450 nm was measured.

### 2.6. Virus Neutralization Test (VNT)

A cytopathic effect-based virus neutralization test (CPE-VNT100) was performed to investigate antibodies capable of neutralizing SARS-CoV-2 in a biosafety level 3 laboratory, following WHO recommendations [38]. 

The experiment was carried out with two variants of SARS-CoV-2: Variant B (MT350282) and Delta (EPI_ISL_2965577). Vero cells (ATCC CCL-81) were seeded in 96-well plates containing 5 × 10^4^ cells/mL 24 h before the infection. Eight serum dilutions were performed from 1:20 to 1:2560. Likewise, 100 TCID50 of the virus was added to all wells, except for the negative control, which was a culture medium without the virus. The serum and virus mixture were incubated at 37 °C under 5% CO_2_ for 1 h to allow virus neutralization and were then transferred onto the confluent cell monolayer and incubated for 72 h under the same conditions. After three days, the plates were analyzed with an Invitrogen™ EVOS™ M5000 Imaging System microscope, and each well with cells was classified with the absence or presence of a cytopathic effect caused by SARS. The value of the virus neutralization titer (1:20 to 1:2560) was defined according to the highest dilution of the serum that neutralized the virus growth. The plates were fixed and stained with 0.2% Naphthol Blue Black solution (Sigma-Aldrich Co., Deisenhofen, Germany) for 30 min. For each assay, an internal positive and negative control serum were used. This method was adapted from Nurtop et al. [39], and similar methods were used for other serological studies of SARS-CoV-2 [40,41,42,43].

### 2.7. Determination of Specific IgM and IgG for CMV

The seropositivity for CMV was evaluated through the assessment of the plasma level of specific IgM and IgG for CMV by using commercial ELISA test kits (BioClin, MG, Brazil). The CMV seropositivity of the volunteers has been defined as an IgM and IgG concentration ≥1.1 and ≥1.32 IU/mL, respectively, in agreement with the manufacturer’s instructions.

### 2.8. Genotyping for the Polymorphism in the IFN-Lambda (il28b Gene)

The polymorphism in IFN-lambda was carried out through the il28b gene, which is known as a member of a family encoding IFN-lambda-like cytokine (rs12979860 C > T). We applied the TaqMan assay allelic discrimination method for single nucleotide polymorphism (SNP) genotyping (Applied Biosystems, Invitrogen, MA, USA), following the description of Prokunina-Olsson et al. [44].

Briefly, the PBMCs, previously stored, were submitted to thawing, and soon after they were subjected to DNA extraction using the PureLink^®^ Genomic DNA Mini Kit, in accordance with the manufacturer’s instructions (Thermo Fisher Scientific, Invitrogen, MA, USA). The extracted DNA was quantified in a Thermo Scientific™ NanoDrop™ One Microvolume UV-Vis spectrophotometer in order to verify the efficiency of the extraction and also to estimate the quality of the DNA obtained. Then, about 4 ng of DNA from each sample was used for amplification in RT-PCR through the TaqMan system with probes, primers, and cycling conditions following a previously published protocol [44]. The reaction was performed on the StepOnePlus™ Real-Time PCR System Platform with a computer attached.

### 2.9. Statistical Analysis

The results were initially evaluated for normality using the Shapiro–Wilk test, and the homogeneity of variance was evaluated using the Levene test.

Due to the fact that all data were considered as non-parametric variables, the results are represented as the median and interquartile intervals. In order to analyze whether there were significant differences between the results obtained in the present study, we used the Wilcoxon test and Kruskal–Wallis test with the Müller–Dunn post-hoc test. In addition, the Spearman correlation coefficient test was also used. Clinical characteristics and symptoms were evaluated using the Chi–square test.

All the analyses were performed with GraphPad Prism 8.1.2 software and the significance level was defined as *p* < 0.05.

## 3. Results

### 3.1. Sample Characterization

Table 1 presents the volunteers’ anthropometric data and clinical symptoms presented by the volunteers during the MO period. Whilst the male group showed greater weight and height than the women, the BMI was similar between them.

### 3.2. Specific IgG for the CMV and SARS-CoV-2 Antigens

Figure 1 shows the results obtained in the evaluation of specific IgG for SARS-CoV-2 (Figure 1A,B) and also for CMV (Figure 1C). As expected, the levels of specific IgG for SARS-CoV-2 were significantly greater in the MO (*p* < 0.01) and VAC (*p* < 0.001) time-points as compared to the values found in the CURE time-point (Figure 1A). Beyond these findings, it is worth mentioning that 17 volunteers (68%) presented elevation in the specific IgG levels for SARS-CoV-2 after the vaccination with the ChadOx-1 vaccine (Figure 1B). Concerning the results of specific IgG levels for CMV, interestingly, these levels were significantly greater in the CURE (*p* < 0.05) and VAC (*p* < 0.001) time-points as compared to the values observed in the MO time-point (Figure 1C). In addition, the specific IgG levels for CMV in the VAC time-point were also higher than in the MO time-point (*p* < 0.01, Figure 1C). No volunteers presented specific IgM for CMV antigens.

In addition to these analyses, we also performed the cytopathic effect-based virus neutralization test for SARS-CoV-2, and we observed that only three volunteers (12%) presented neutralizing antibodies at the MO time-point (ranging between the titers 20 and 640), whereas at the CURE time-point nine volunteers (36%) presented neutralizing antibodies (ranging between 40–320), and at the VAC time-point twelve volunteers (48%) presented neutralizing antibodies (ranging between 20 and >2560). The results concerning the individual neutralizing titers for both variants are shown in the Appendix A. Based on these observations, and in accordance with our previous study [45], in which the presence of neutralizing antibodies was used to separate the volunteers into responder and non-responder groups to influenza virus vaccination, Figure 1D shows that the responder group (composed of the volunteers who presented neutralizing antibodies) showed higher specific IgG levels for COVID-19 than the non-responder group, regardless of the time-point assessed, in contrast to the lower specific IgG levels for CMV found in the responder group compared to the non-responder group (Figure 1E).

Since gender can impact the immune response to infections and vaccination, Figure 2 shows the specific IgG for SARS-CoV-2 (Figure 2A) and CMV (Figure 2B) antigens in the volunteer group who participated in this present study separated into older-men and older-women subgroups. In the intra-subgroup analysis, it was observed that the older-men group showed a significant reduction of specific IgG levels for SARS-CoV-2 antigens (Figure 2A) in the CURE time-point as compared to values in the MO time-point (*p* = 0.0304), followed by an increase in these levels in the VAC time-point in relation to the values found in the CURE time-point (*p* = 0.0067). Concerning the specific IgG levels for CMV antigens (Figure 2B), the older-women subgroup showed a significant increase in these levels only in the VAC time-point in comparison to values observed in the MO time-point (*p* = 0.0335), whereas the older-men subgroup showed not only a significant increase in these levels in the CURE (*p* = 0.0269) and VAC (*p* = 0.0002) time-points in comparison to values observed in the MO time-point, but also in the CURE time-point in relation to the MO time-point (*p* = 0.0102). Regarding the inter-subgroup analysis, no significant differences were found between the values observed in the older-men and older-women subgroups for either the SARS-CoV-2 (Figure 2A) or CMV antigens (Figure 2B).

Based on the observation that there were no differences in the specific IgG response for SARS-CoV-2 and CMV antigens between the older-men and older-women subgroups, we have decided to present the next results obtained in these subgroups together.

Figure 3 presents the results obtained in the analysis of the Spearman correlation coefficient between the values of specific IgG for SARS-CoV-2 and CMV in the three time-points studied here. A significant negative correlation in the MO time-point was observed (Figure 3A), whereas in other time-points no significant differences were found (CURE–Figure 3B and VAC–Figure 3C).

### 3.3. Allelic Discrimination of the Genotyping for the Polymorphism in IFN-Lambda (il28b Gene)

Figure 4 shows the allelic discrimination plot obtained after the genotyping assay for the polymorphism in the il-28b gene (rs12979860), an IFN-lambda-like cytokine. In red are shown the volunteers characterized as homozygous for allele 1 (C/C, n = 9, 36%). In blue are shown the volunteers characterized as homozygous for allele 2 (T/T, n = 4, 16%). In green are shown the volunteers characterized as heterozygous for alleles 1 and 2 (C/T, n = 12, 48%). It is important to mention that four volunteers in the Allele 1 subgroup (44,45%) presented moderate or mild symptoms, whereas only one volunteer in the Allele 2 subgroup (25%) and two volunteers in the Allele 1 and 2 subgroup (16%) presented only mild symptoms.

### 3.4. Specific IgG for the CMV and SARS-CoV-2 Antigens in the Volunteers Grouped Based on the Allelic Discrimination

Figure 5 shows the results obtained in the evaluation of specific IgG levels for SARS-CoV-2 and CMV, as well as these levels in the responder (R) and non-responder (NR) groups in the three time-points studied here when the volunteers were grouped in their respective polymorphisms. Figure 5A shows that only the levels of IgG in the CURE time-point were higher than the MO time-point in the volunteers who presented homozygous for allele 1. No other differences were found both for the assessment of specific IgG levels for SARS-CoV-2 (Figure 4) and CMV (Figure 5C). Concerning the results observed in the specific IgG levels for SARS-CoV-2 (Figure 5B) and CMV (Figure 5D) in the responder and non-responder groups, it was found that the specific IgG levels for SARS-CoV-2 in the responder’s group were higher at CURE and VAC time-points than the non-responder´s group in the Allele 1 subgroup, whereas in the Allele 1 and 2 subgroup higher specific IgG levels for SARS-CoV-2 were found in the responder’s group compared to the non-responder´s group at MO, CURE, and VAC time-points (Figure 5B). In contrast, the specific IgG levels for CMV were lower in the responder´s group than non-responder´s group both in the Allele 1 subgroup at CURE and VAC time-points, and in the Allele 1 and 2 subgroups at MO, CURE, and VAC time-points (Figure 5D). It is noteworthy to mention that the significant differences observed between non-responder and responder groups at the MO time-point in the Allele 1 subgroup, in Figure 5C,D, are closely related to the fact that no volunteer presented neutralizing antibodies assessed in the VNT.

In Figure 6, the results obtained in the Spearman correlation coefficient analysis between the specific IgG levels for SARS-CoV-2 and CMV in the volunteers separated into groups are shown as follows: homozygous for allele 1 (Figure 6A,D,G); homozygous for allele 2 (Figure 6B,E,H); and heterozygous for alleles 1 and 2 (Figure 6C,F,I) in the MO, CURE, and VAC time-points. Significant negative correlations were found in the volunteers who presented homozygous for allele 1 in the MO time-point (Figure 6A) and also in the volunteers who presented heterozygous in the CURE time-point. It is noteworthy to mention that in the CURE time-point, the results observed in the same group showed a relevant tendency (*p* = 0.0670) of negative correlation between these specific IgG levels (Figure 6D). No other significant correlations were found.

## 4. Discussion

In the present study, our results showed that both SARS-CoV-2 infection and COVID-19 vaccination elicited a robust immune response in older adults, not only in terms of the significant increase in serum-specific IgG levels for SARS-CoV-2 antigens, but also in terms of immunogenicity (more than 65%). Furthermore, we observed that the specific serum IgG levels for CMV significantly increased over time. Of interest, these results were not impacted by gender. Regarding the impact of seropositivity for CMV in the COVID-19 context, we observed a significant negative correlation between specific IgG levels for SARS-CoV-2 and CMV at time-point MO in the volunteer group. Interestingly, this finding could be putatively associated with polymorphism for IFN-lambda, mainly in the homozygous for allele 1 group (C/C), since we found a significant negative correlation between the levels of specific IgG for SARS-CoV-2 and CMV in individuals who presented homozygous for this allele 1 in the MO time-point and heterozygotes in the CURE time-point. Moreover, it is worth highlighting that the responder´s group showed a better response in terms of specific IgG levels for SARS-CoV-2, and this finding could be associated with the reduced specific IgG levels for CMV observed in the same volunteer group.

Although several studies have been developed aiming to amplify our understanding of how immune responses are elicited and respond to both SARS-CoV-2 infection as well as to COVID-19 vaccination, particularly in the older adult population, the impact of some factors that undoubtedly can interfere with the induction of an efficient immune response against viruses, such as chronic CMV infection and polymorphism in the lambda interferon gene, still need to be elucidated.

Among several immune senescence-related aspects, one of the most important concerns is the reduced vaccination response since, as formerly mentioned, the vaccine immunogenicity against the influenza virus in the older adult population was around only 30–40% [46], which was clearly low compared to younger/adult populations that were around 70–90% [13]. At this point, it is important to point out that vaccine immunogenicity can be defined as “the strength or magnitude of an immune response”, according to Smetana et al. [46].

Based on these pieces of information, differing from the immunogenicity for the influenza virus vaccination, our findings concerning the antibody response, particularly specific IgG, in the volunteers submitted to COVID-9 vaccination with ChadOx-1, showed not only a significant increase in their circulating levels post-vaccination, but also in immunogenicity above that expected for the elderly population (~65%). These results allow us to suggest that the ChadOx-1 vaccine presents a prominent potential to elicit a robust humoral immune response in the older-adult population, which, in the context of this study, can be closely associated with the activation of specific immunological memory for COVID-19, since all the volunteers were previously infected with SARS-CoV-2.

These results can raise our expectations concerning the capacity of the vaccines for COVID-19 to effectively prevent at least the occurrence of severe COVID-19 cases, mainly in the older-adult population. The impacts of chronic CMV infection on the response to vaccination in this population is still controversial. In accordance with the literature, even if most of the older adults are CMV-seropositive, and it has been demonstrated that the chronic presence of this can negatively impact the specific antibody responses to the vaccine against the influenza virus [47,48,49,50], other studies showed no effect [48,51,52,53] Therefore, it is still debated whether this long-lasting infection can really impair immune response to vaccinations [54].

Regarding the findings on specific IgM and IgG levels for CMV antigens, the observation that none of the volunteers presented detected IgM levels allows us to suggest that all of them were chronically infected with CMV. Furthermore, the significant elevation of specific IgG levels for CMV along with the time is very interesting, and can allow us to putatively suggest that the SARS-CoV-2 infection could have altered the systemic inflammatory status favoring the inflammaging, thus generating a favorable environment for CMV reactivation [21], which impacted the immune response and led to an increase in specific IgG levels for CMV. This suggestion can be supported by the recent reports that demonstrate the potential of SARS-CoV-2 infection to promote the reactivation of β-herpesviruses, such as CMV, which can cause opportunistic infection, including in patients with COVID-19 (mainly those who are critically ill), due to some epiphenomena related to the severity of COVID-19, such as intensive care unit length-of-stay, inflammation, and/ or treatment with corticosteroids [55,56,57]. Although these observations are interesting, it is important to mention that the increase in specific IgG levels for CMV over time did not significantly interfere with the humoral immune response to the COVID-19 vaccine, since there was no correlation between specific IgG levels for COVID-19 and CMV at VAC time-point.

However, in a different way, the analysis of the correlation between the specific IgG levels for these viruses at the MO time-point showed a significant negative correlation, indicating that, in general, those older adults with better specific IgG response to SARS-CoV-2 presented lower levels of specific IgG for CMV. Despite that it has been cited that CMV serostatus does not directly impact the response to vaccination magnitude [58], our finding corroborates some reports that point to the fact that CMV chronic infection status, which could be assessed by both humoral and cellular immune responses, can significantly interfere with the immune response to infection with other viral infectious agents [59], such as the influenza virus [51], or even SARS-CoV-2, in the older adult population, particularly due to immunosenescence and inflammaging [60]. In fact, our results observed in the responder´s group can corroborate these pieces of information, since those volunteers presented better specific IgG levels for SARS-CoV-2 in association with the lower specific IgG levels for CMV.

Considering that CMV seropositivity was negatively correlated with specific IgG response for COVID-19, specifically at MO time-point, we decided to investigate whether the polymorphism in the gene for interferon lambda (il28b) could be an important piece in this puzzle. As previously mentioned, IFN-lambda is one of the most important cytokines for establishing viral resistance and activating adaptive immune responses against viruses, including CMV and SARS-CoV-2 [31,61,62].

Regarding the immune response elicited by SARS-CoV-2 infection, locally in the upper airway mucosa, the well-known site of initiation of this infection, it was verified that IFN-lambda presented a higher capacity in restricting viral spread from the nasal epithelium to the respiratory tract than type I interferons [63]. Although the expression of type I interferon receptors is ubiquitous, the expression of type III interferon receptors is limited to tissues with relatively high numbers of epithelial cells, such as the lungs, skin, and gastrointestinal and respiratory tracts. So, this pattern of receptor distribution favors the type III interferon family, which includes IFN-lambda, to create a robust immune response in these tissues against viruses [64,65].

Since IFN-lambda has remarkable action on viruses, which can include CMV and SARS-CoV-2 infection, we evaluated the single nucleotide polymorphism (SNP) for the IL-28 gene (rs12979860 C > T), an interferon-like cytokine, specifically of type III or lambda-3 [65,66,67]. In this respect, the SNP evaluated was associated with the presence of only cytosine in the pair of alleles (C/C), classified as homozygous for allele 1, or only of the thymine in the pair of alleles (T/T), classified as homozygous for allele 2, or heterozygous (C/T), in the 3Kpb upstream region of the il28b gene [68]. Based on this description, it was possible to verify that nine (36%) volunteers were homozygous for allele 1, four (16%) were homozygous for allele 2, and twelve (48%) were heterozygous.

After these findings, we assessed the influence of the presence of those homozygous for allele 1 or allele 2, or heterozygous in the specific IgG response for CMV and SARS-CoV-2, in our volunteer group in all time points evaluated in this study. Although the presence of those homozygous for allele 2 (T/T) or heterozygous (C/T) did not impact the specific IgG responses, both in the context of COVID-19 and CMV, at the different time points studied here, a significant increase in specific IgG levels to SARS-CoV-2 antigens was found in the CURE time-point as compared to the values found in the MO time-point exclusively in the volunteer group that presented homozygous for allele 1 (C/C). This result can allow us to suggest that the presence of those homozygous for allele 1 led to a delay in the production of specific IgG to SARS-CoV-2 infection, and also that the IgG levels remained high eight months after the MO time-point. Corroborating these findings, we were able to demonstrate that the volunteers in the Allele 1 subgroup not only did not show neutralizing antibodies at the MO time-point, but also showed the greatest increase in the specific IgG levels for SARS-CoV-2 at the CURE time-point, specifically in the responder´s group.

The association between the polymorphism in the IFN-lambda gene (rs12979860) and COVID-19 has been reported in the literature. In this sense, Agwa and collaborators (2021) documented that COVID-19 patients showed a higher frequency of allele 1 (C/C) than healthy controls. In the same report, the authors also observed that the disease was more severe in patients who presented allele 1 or who were heterozygous than those presenting allele 2 (T/T). Furthermore, it is important to mention that the authors also reported that individuals who were heterozygous (C/T) presented a poor prognosis for COVID-19 than other genotypes [68]. These reports can partially corroborate our findings since the Allele 1 subgroup presented a higher number of individuals with symptoms (four of nine; 44,45%), including two volunteers with moderate symptoms, even though the Allele 1 and 2 subgroup presented a minor number of symptomatic volunteers. On the other hand, Saponi-Cortes et al. [69] reported that allele 2 (TT), in the IFN-lambda gene (rs12979860), was significantly overexpressed in COVID-19 patients compared to non-COVID-19 controls [69]. According to the same authors, and with some information previously cited, whereas the T allele has been associated with an ineffective viral clearance, e.g., for HCV and other RNA viruses, the presence of the C allele could favor the clearance of these viruses [28,29,69,70]. Although it has been emphasized that some genetic polymorphisms, particularly in the IFN-lambda gene, could be involved in susceptibility to COVID-19 [71], the discrepant results found until now in the literature reinforce the need for further studies focused on improving our understanding of the impact of polymorphism in the IFN-lambda gene in the COVID-19 context.

Even though we cannot affirm the following, a suggested explanation for the delayed elevation of the specific IgG levels for COVID-19 in the volunteers who presented allele 1 (C/C) could be associated with CMV seropositivity, since in this subgroup of volunteers we found a negative correlation between the specific IgG levels for SARS-CoV-2 and CMV at the MO time-point. These data corroborate the literature, since it has been reported that individuals with genotypes CT or TT presented better control of CMV replication following primary infection than individuals presenting genotype CC, due to the upregulation of interferon-sensitive genes (ISGs), which acts in the inhibition of viral replication [71,72]. So, this result allows us to putatively suggest that IFN-lambda polymorphism in CMV-seropositive patients can impact response to SARS-CoV-2 infection. Moreover, it is noteworthy to point out that during the CURE time-point, whilst a tendency to perpetuate this negative influence of CMV seropositivity on the response to SARS-CoV-2 infection was observed, exclusively in the subgroup of older adults who presented allele 1 (C/C), the subgroup with older adults who were heterozygous (C/T) presented a negative correlation between specific IgG levels for CMV and SARS-CoV-2. Beyond these findings, we also showed that, in a general way, the responder groups, regardless of polymorphism types, presented a better response in the specific IgG levels for SARS-CoV-2 in association with lower specific IgG levels for CMV. It is of utmost importance to point out that although the results found in the Allele 2 subgroup were similar to those observed in other subgroups, the low number of volunteers in that group precluded a statistical analysis.

In fact, according to the literature, it has been reported that the presence of allele 2 (T/T) is associated not only with lower replication permission and incidence of active CMV infection, but also with higher gene expression for IFNs than the presence of allele 1 (C/C) and heterozygous patients [69,72]. Furthermore, Bravo and collaborators [68] showed that patients undergoing allogeneic stem cell transplantation with allele 2 (T/T) in the IFN-lambda-3 gene showed a shorter duration of the first episodes of CMV infection than patients with allele 1 (C/C) and heterozygous patients (C/T). Considering these pieces of information, it is reasonable to suppose that the presence of allele 2 (T/T) can drive more protective action against CMV infection. Therefore, we may presumably suggest that the occurrence of those homozygous for allele 1 (C/C), by generating an environment more favorable to CMV, may interfere with immune response against other viruses, such as SARS-CoV-2, which was observed in this study.

## 5. Limitations of the Study

The limitations of the study were: (1) The discrepant number of older men and older women enrolled in this study. In this respect, it is important to mention that this difference could be attributed to the fact that these volunteers were exclusively recruited from the older adults who lived in the “Hospital Geriátrico e de Convalescentes Dom Pedro II”, an institution in which aged people can live for a long period, similar to an asylum. Although, the data presented in Figure 2 can minimize this drawback. (2) The lack of a group of adults and/or young adults with equivalent SARS-CoV-2 infection, COVID-19 immunization, and CMV status, which could not only improve our understanding of the immunogenicity of this vaccine in the older-adult population, but also verify whether the CMV seropositivity and IFN-lambda polymorphism could affect the immune response to COVID-19 in populations with another age group. (3) The lack of a group of older adults presenting mono-infection for CMV and SARS-CoV-2, which could be used as a control in order to evaluate the specific IgG and neutralizing antibodies in the three groups analyzed in the present study. (4) The lack of data regarding the presence or absence of active CMV infection in volunteers at different time points evaluated in this study. (5) The lack of evaluation concerning the systemic levels of cytokines (both pro- and anti-inflammatory profiles), which could improve our understanding of the inflammaging status of our volunteers.

## 6. Conclusions

Based on the results obtained in this study, it is possible to conclude that: (1) Remarkable immunogenicity for SARS-CoV-2 antigens was present in the older adults vaccinated with ChadOx-1; (2) CMV seropositivity negatively impacted the specific IgG response to COVID-19; and (3) The heterozygosis (C/T) and mainly the homozygosis for allele 1 (C/C) in the il28b gene, which encodes an IFN-lambda-3 cytokine, could putatively be an important factor to be assessed in the specific antibody response to COVID-19 in older adults seropositive for CMV.

## Figures and Tables

**Figure 1 vaccines-11-00480-f001:**
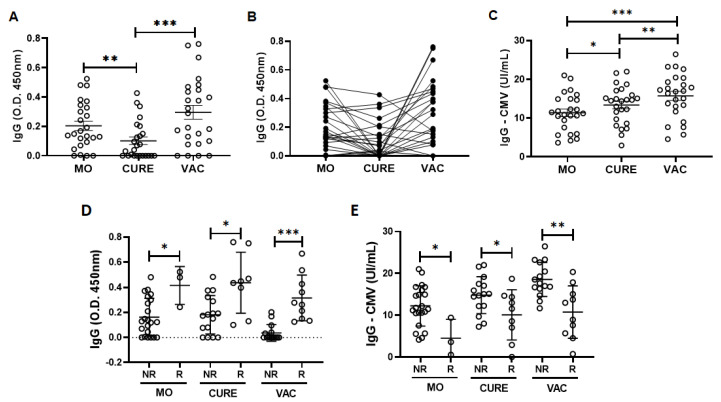
Total serum levels of IgG specific for SARS-CoV-2 (O.D. 450 nm—(**A**)) and CMV (UI/mL—(**C**)) antigens, and the dynamics of serum IgG levels for SARS-CoV-2 (**B**), in a group of older adults who participated in the study on three different occasions: during the micro-outbreak of SARS-CoV-2 infection (MO), 6–8 months after this infection (CURE), and 30 days after administration of the second dose of ChadOx-1 vaccine (VAC). Also presented here are the total serum levels of IgG specific for SARS-CoV-2 (O.D. 450 nm—(**D**)) and CMV (O.D. 450 nm—(**E**)) antigens when the volunteers were separated in accordance with the presence (R—responders) and not (NR—non-responders) of neutralizing antibodies for SARS-CoV-2. * denotes values of *p* < 0.05; ** *p* < 0.01; *** *p* < 0.001.

**Figure 2 vaccines-11-00480-f002:**
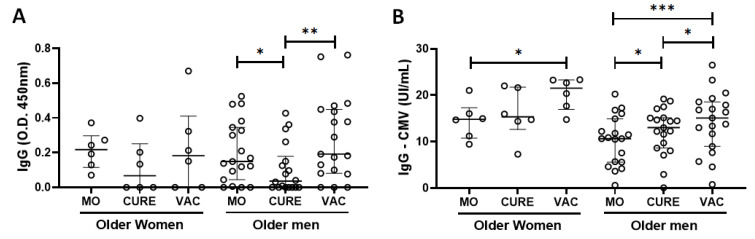
Total serum levels of IgG specific for SARS-CoV-2 (O.D. 450 nm—(**A**)) and CMV (UI/mL—(**B**)) antigens in the group of older adults who participated in the study separated in the subgroups of “men” and “women” on three different occasions: during the micro-outbreak of SARS-CoV-2 infection (MO), 6–8 months after this infection (CURE), and 30 days after administration of the second dose of ChadOx-1 vaccine (VAC). * denotes values of *p* < 0.05; ** *p* < 0.01; *** *p* < 0.001.

**Figure 3 vaccines-11-00480-f003:**
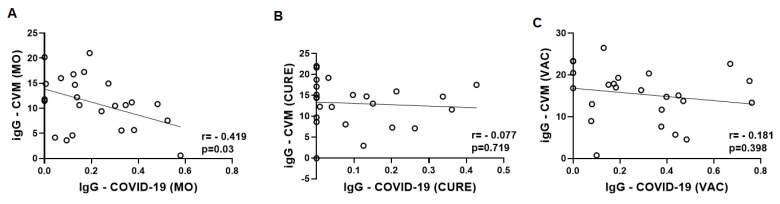
Spearman coefficient correlation analysis between total serum levels of CMV-specific IgG (IU/mL) and COVID-19-specific IgG (O.D. 450 nm) in the older adults who participated in the study on three different occasions: during the micro-outbreak of SARS-CoV-2 infection (MO, (**A**)), 6–8 months after this infection (CURE, (**B**)), and 30 days after administration of the second dose of ChadOx-1 vaccine (VAC, (**C**)).

**Figure 4 vaccines-11-00480-f004:**
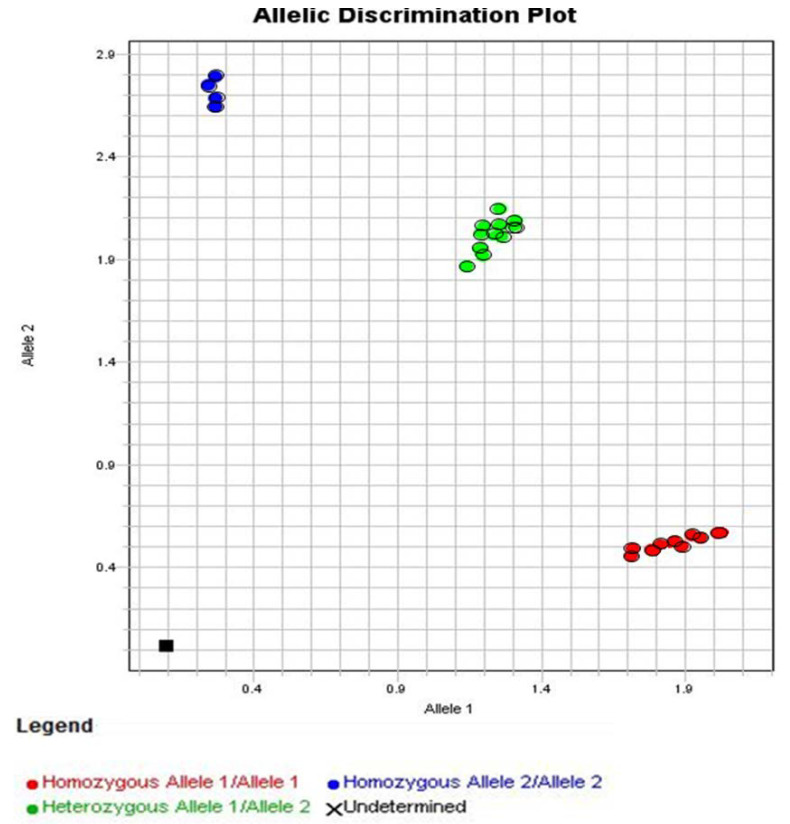
Representative graphic of allelic discrimination obtained after the genotyping assay for polymorphism in the il28b gene (rs12979860) in the older adults who participated in the study. In red color are presented the volunteers homozygous for allele 1, which corresponds to the nitrogenous base cytosine. In blue are presented the volunteers homozygous for allele 2, corresponding to the nitrogenous base thymine. In green are presented the heterozygous volunteers.

**Figure 5 vaccines-11-00480-f005:**
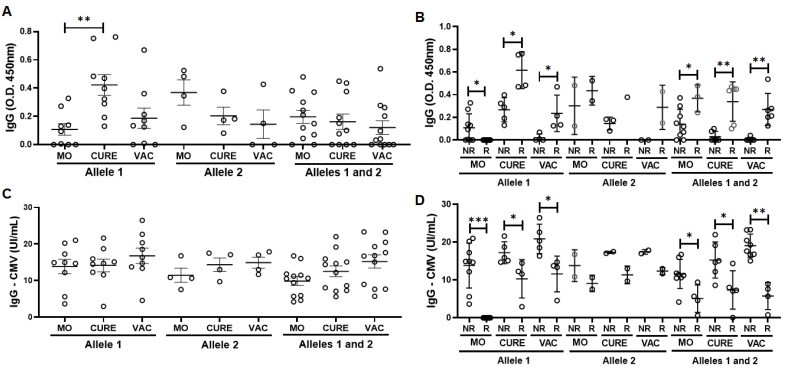
Total serum levels of IgG specific for SARS-CoV-2 (O.D. 450 nm, (**A**)) and CMV (UI/mL, (**C**)) antigens in the subgroups of volunteers grouped in accordance with the polymorphism in the IFN-lambda gene [allele 1 (C/C), allele 2 (T/T), and alleles 1 and 2 (heterozygous, C/T)], as well as these levels in the same subgroups of volunteers separated in accordance with the presence (R—responders) or not (NR—non-responders) of neutralizing antibodies for SARS-CoV-2, respectively (SARS-CoV-2—(**B**) and CMV—(**D**)). * denotes values of *p* < 0.05; ** *p* < 0.01; *** *p* < 0.001.

**Figure 6 vaccines-11-00480-f006:**
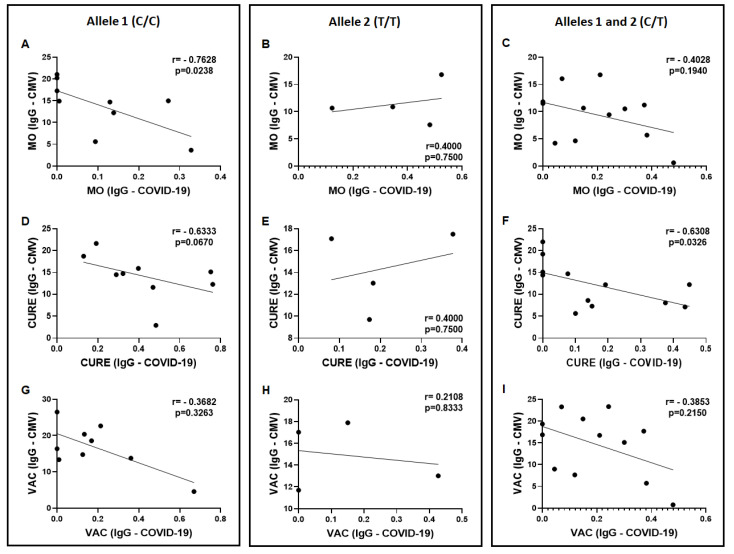
Spearman coefficient correlation analysis between total serum levels of CMV-specific IgG (IU/mL) and COVID-19 IgG (O.D. 450 nm) in the volunteers grouped in accordance with the polymorphism in the IFN-lambda gene [allele 1 (C/C), allele 2 (T/T), and alleles 1 and 2 (heterozygous, C/T)], on three different occasions: during the micro-outbreak of SARS-CoV-2 infection (MO—(**A**–**C**), respectively), 6–8 months after this infection (CURE—(**D**–**F**), respectively), and 30 days after administration of the second dose of ChadOx-1 vaccine (VAC—(**G**–**I**), respectively).

**Table 1 vaccines-11-00480-t001:** Anthropometric data presented as mean and standard deviation (X_±_SD), and also the clinical characteristics (n) presented by the volunteers during the MO period.

Variables		Volunteers		*p*-Value
	Total	Women	Men	
	(n = 25)	(n = 06)	(n = 19)	
Age (years)	75.8 ± 8.5	75.4 ± 8.4	77.0 ± 9.7	0.483
Height (m)	1.68 ± 0.09	1.58 ± 0.07 *	1.71 ± 0.08	**0.003**
Weight (kg)	75.2 ± 10.9	63.3 ± 6.8 *	79.1 ± 9.1	**0.001**
BMI (kg/m^2^)	26.5 ± 2.9	25.9 ± 2.5	26.9 ± 3.1	0.199
Clinical characteristics (n)				
Cough	5	1	4	>0.05
Fever	3	0	3	>0.05
Sore throat	1	0	1	>0.05
Coryza	2	0	2	>0.05
Oxygen Desaturation	2	1	1	>0.05
Respiratory distress	2	1	1	>0.05

* significant difference between the women and men groups.

## Data Availability

Data availability is under the responsibility of the authors and any data can be made available upon request and need.

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
