# Peer review of "Assessment of the Interferon-Lambda-3 Polymorphism in the Antibody Response to COVID-19 in Older Adults Seropositive for CMV"

_vaccines, 2023, doi:10.3390/vaccines11020480_

Round 1

Reviewer 1 Report

Authors followed six women and 19 men in the age of 51-85 years old, who got COVID19 and hospitalized, recovered and received twice of ChadOx-1 (AstraZeneca/Oxford) vaccine. Blood samples were collected on three different occasions: during a micro-outbreak of COVID-19 in the hospital (MO), eight months after this micro-outbreak (CURE), and 30 days after the administration of the second dose of vaccine (VAC).  Antibody titre against COVID-19 and CMV were evaluated. Also neutralizing antibodies against COVID 19 infection were examined for, denominated as responders and non-responders. Furthermore, IFN lambda sub-type were measured; type I(C/C), typeI/II(C/T), type II (T/T).   Authors showed that anti SARS-CoV2 antibodies increased in MO and VAC, decreased in CURE (levels of antibodies CURE<MO<VAC). Anti CMV antibodies was MO<CURE<VAC. In this study population, IFN lambda type II homologous population was n=4. Type I homogenous or type I/II heterogenous population had more infected cases and in these populations, CMV seropositivity was negatively correlated with specific IgG response for COVID-19, specifically at MO time-point.

It is an interesting article and results were shown clearly. However, as authors mentioned sample numbers is limited and therefore some of statistical analysis could not be trusted. It would be recommended to add more number of patients, if possible.

1.     In order to improve readability, I will suggest to send English correction. There are several typos in introduction and materials and methods. Please correct.

a.      Line 57-60, please correct the underlined part.

Particularly in terms of B cells activity, it was reported that neutralizing antibodies (IgM and IgG) response after Influenza virus vaccination in older adults showed was significantly decreased not only by their lower production but also by their shorter half-life and low affinity for antigens [11].

b.      Line 141, 179, 192, 232: add space between numbers and unit

c.      Line 145, 146, 162, 167, 200, 203: correct the size of letters

d.      Line 243: posthoc should be post-hoc

2.      Authors stated older population when describe the cited references. In this article, volunteer age was 51-85 years old. Most of references in elderly population is >65 years old. This age fact could be important for the results. Therefore, if the cited references have different age definitions compared to this manuscript, please indicate.

3.      Line 61; efficacy of influenza vaccine to produce antibodies were 30-40% in elderly population. How much higher in younger population? And what is the age limit of the groups? Please add this information.

4.      Table 1: please explain * stated in the hight of women population. Calculation value for hight

Height (m)  total (n=25) 1,57±0,07 women (n=6) 1,53±0,08* men (n=19) 1,70±0,09

Mean value of total hight and SD seems to be incorrect. Please check if this is correctly shown.

5.      Line 270-273, authors explained well antibody levels and neutralizing antibody positivity. Supplemental Table is missing in the manuscript which I got. If possible, please send the attachment separately.

6.      Figure 1 D and E: please add the numbers of each group or indicate individual dots in a similar way as in Figure 1 A and B.

7.      Figure 5 B and D: show individual dots or indicate numbers in each group. If it gets too crowded in figures, data of how many are responder and non-responder in association to Figure 5 should be shown in a Table.

8.      Line 417: Should be changed as: Among several immune senescence-related aspects,

9.      Line 418-422 is not easy to understand. The sentence could be rephrased as below:

Smetana et al. described that vaccine immunogenicity is defined as “the strength or magnitude of an immune response.” [40]. In the same report, it was shown that the vaccine immunogenicity against the Influenza virus in the elderly population was around only 30–40%, which was clearly low compared to younger populations (vaccine immunogenicity in younger population was ….

10.   Line 447  ‘herpesviruses, such as CMV,’ this is not incorrect but could lead to misreading. This should be ‘Herpesviridae, such as CMV,’ or 'β-herpesviruses, such as CMV, which can cause opportunistic infection,’

11.   Line 458-459  ‘monitoring of CMV IgG levels is important for patient follow-up by reflecting the infection status’

It will be helpful for reader to understand the point if authors can explain how CMV IgG levels reflect the infectious status in this report.

12.   Line 511, ‘By the way’ could be replaced to ‘Furthermore’ or ‘In addition,’.

13.   Line 516-517 'On the other hand, it was reported that the presence of allele 2 in IFN-lambda-3 gene (rs12979860) was related to more incidence of COVID-19 in Spain [65]. '

 Please speculate the reasons of these controversial data and discuss, which can suggest more direction of further study.

Author Response

Reviewer #1

Comments and Suggestions for Authors

Authors followed six women and 19 men in the age of 51-85 years old, who got COVID19 and hospitalized, recovered and received twice of ChadOx-1 (AstraZeneca/Oxford) vaccine. Blood samples were collected on three different occasions: during a micro-outbreak of COVID-19 in the hospital (MO), eight months after this micro-outbreak (CURE), and 30 days after the administration of the second dose of vaccine (VAC).  Antibody titre against COVID-19 and CMV were evaluated. Also neutralizing antibodies against COVID 19 infection were examined for, denominated as responders and non-responders. Furthermore, IFN lambda sub-type were measured; type I(C/C), typeI/II(C/T), type II (T/T).   Authors showed that anti SARS-CoV2 antibodies increased in MO and VAC, decreased in CURE (levels of antibodies CURE<MO<VAC). Anti CMV antibodies was MO<CURE<VAC. In this study population, IFN lambda type II homologous population was n=4. Type I homogenous or type I/II heterogenous population had more infected cases and in these populations, CMV seropositivity was negatively correlated with specific IgG response for COVID-19, specifically at MO time-point.

It is an interesting article and results were shown clearly. However, as authors mentioned sample numbers is limited and therefore some of statistical analysis could not be trusted. It would be recommended to add more number of patients, if possible.

- Authors comments: First of all, we sincerely would like to thank you for the criticism and also for your constructive comments/suggestions that enable us to improve this study. In addition, it is noteworthy to clarify that all alterations concerning your comments/suggestions in the main text are marked in red.

Regarding the recommendation to add more volunteers to the present study, we would like to mention that, unfortunately, we cannot do so, because: (1) all the volunteers who participated in this study were recruited exclusively among the older adults who lived in the "Hospital Geriátrico e de Convalescentes Dom Pedro II", an institution in which the aged people can live for a long period, similarly to an asylum, then, all of them were submitted to the same conditions, (2) all of them were infected with SARS-CoV-2 on the same occasion, in 2020, and continued to reside in this hospital until the last blood sampling collection after administration of the second dose of the vaccine for the COVID-19, and (3) no other cases of individuals infected with SARS-CoV-2 were reported (asymptomatic cases), therefore, no blood sample was collected in the same period of development of the study.

  1.    In order to improve readability, I will suggest to send English correction. There are several typos in introduction and materials and methods. Please correct.

- Authors´ comments: We would like to thank you for the comment and to inform you that, as recommended, a revision of the English (spelling and grammar) was performed in order to improve its readability. 

  1. Line 57-60, please correct the underlined part.

Particularly in terms of B cells activity, it was reported that neutralizing antibodies (IgM and IgG) response after Influenza virus vaccination in older adults showed was significantly decreased not only by their lower production but also by their shorter half-life and low affinity for antigens [11].

- Authors´ comments: We would like to thank you for the comment and inform you that this sentence was revised, as presented below.

"Particularly in terms of B cells activity, it was reported that neutralizing antibodies (IgM and IgG) response after Influenza virus vaccination in older adults showed a significant decrease not only by their lower production but also by their shorter half-life and low affinity for antigens [11]."

  1. Line 141, 179, 192, 232: add space between numbers and unit

- Authors´ comments: We would like to thank you for the comment and, as recommended, space between numbers and units were added.

  1. Line 145, 146, 162, 167, 200, 203: correct the size of letters

- Authors´ comments: We would like to thank you for the comment and, as recommended, we revised the text and corrected the size of the letters.

  1. Line 243: posthoc should be post-hoc

- Authors´ comments: We would like to thank you for the comment and, as recommended, we altered "posthoc" to "post-hoc".

  1. Authors stated older population when describe the cited references. In this article, volunteer age was 51-85 years old. Most of references in elderly population is >65 years old. This age fact could be important for the results. Therefore, if the cited references have different age definitions compared to this manuscript, please indicate.

- Authors´ comments: We would like to thank you for the comment and clarify that although the World Health Organization (WHO) considers that people >65 years old are elderly [Organization WH. WHO. Available from: https://www.who.int/ageing/en/], according to the Health Ministry of Brazil, people >60 years old or more are considered elderly [Brazilian Institute of Geography and Statistics – IBGE. 2010. Available from: line-height:normal"]. Moreover, it is noteworthy to mention that only two volunteers presented with ages below 60 years old. Based on these pieces of information, we opted to use the definition "older adult" in the present study. Lastly, in order to clarify this issue, we added the following sentence in the main text (page XX, lines XXX).

"It is noteworthy to mention that only two volunteers presented with ages below 60 years old."

  1. Line 61; efficacy of influenza vaccine to produce antibodies were 30-40% in elderly population. How much higher in younger population? And what is the age limit of the groups? Please add this information.

- Authors´ comments: We would like to thank you for the comment and clarify that, in agreement with the literature, when the composition of the vaccine coincides with the circulating strain of the virus its effectiveness in healthy adults is between 70% and 90% but falls to between 30% and 50% in individuals over 60 years of age. Besides, it was reported that seasonal influenza vaccine effectiveness in older adults (>65 years old) at about 49%, whereas for healthy younger/adults (18–64 years old) was closer to 59%. So, these pieces of information corroborate our previous report and, as recommended, were added to the main text (page XXX, lines XXX), as presented below.

“In fact, when the influenza vaccine´s composition coincides with the circulating strain of this virus the effectiveness of vaccination in healthy adults is between 70% and 90% but falls to between 30% and 50% in people >60 years old (46). Besides, it was also reported that seasonal influenza vaccine effectiveness in older adults (>65 years old) at about 49%, whereas for healthy younger/adults (18–64 years old) was closer to 59% (14).

  1. Table 1: please explain * stated in the hight of women population. Calculation value for hight

Height (m)  total (n=25) 1,57±0,07 women (n=6) 1,53±0,08* men (n=19) 1,70±0,09

Mean value of total hight and SD seems to be incorrect. Please check if this is correctly shown.

  • Authors´ comments: We would like to thank you for the comment and, as recommended, we revised all the values presented in table 1, and the corrected values were added.

  1. Line 270-273, authors explained well antibody levels and neutralizing antibody positivity. Supplemental Table is missing in the manuscript which I got. If possible, please send the attachment separately.

- Authors´ comments: We would like to thank you for the comment and also to apologize for the Supplementary Table missing. Although we already attached this table to the manuscript submission system, as recommended, we are also sending this table here, for your review.

  1. Figure 1 D and E: please add the numbers of each group or indicate individual dots in a similar way as in Figure 1 A and B.

- Authors´ comments: We would like to thank you for the comment and, as recommended, the presentation of Figures 1D and 1E was revised and the individual dots were added in a similar way to those present in Figures 1A and 1C.

  1. Figure 5 B and D: show individual dots or indicate numbers in each group. If it gets too crowded in figures, data of how many are responder and non-responder in association to Figure 5 should be shown in a Table.

- Authors´ comments: We would like to thank you for the comment and, as recommended, the presentation of Figures 5B and 5D was revised and the individual dots were added in a similar way to those present in Figures 5A and 5C.

  1. Line 417: Should be changed as: Among several immune senescence-related aspects,

- Authors´ comments: We would like to thank you for the comment and inform you that, as recommended, we revised and altered “Among several aspects of immunosenescence-related” to “Among several immune senescence-related aspects”.

  1. Line 418-422 is not easy to understand. The sentence could be rephrased as below:

Smetana et al. described that vaccine immunogenicity is defined as “the strength or magnitude of an immune response.” [40]. In the same report, it was shown that the vaccine immunogenicity against the Influenza virus in the elderly population was around only 30–40%, which was clearly low compared to younger populations (vaccine immunogenicity in younger population was ….

- Authors´ comments: We would like to thank you for the comment and inform you that, as recommended, we revised and altered these sentences as presented below.

" Among several immune senescence-related aspects, one of the most important concerns is the reduced vaccination responses since, as formerly mentioned, the vaccine immunogenicity against the Influenza virus in the older adult population was around only 30-40% [40], which was clearly low compared to younger/adult populations that were around 70-90% [13]. At this point, it is important to point out that vaccine immunogenicity can be defined as “the strength or magnitude of an immune response”, according to Smetana et al [40]."

  1. Line 447  ‘herpesviruses, such as CMV,’ this is not incorrect but could lead to misreading. This should be ‘Herpesviridae, such as CMV,’ or'β-herpesviruses, such as CMV, which can cause opportunistic infection,’

- Authors´ comments: We would like to thank you for the comment and inform you that, as recommended and also in order to avoid misreading, we revised this part and altered it as presented below.

"This suggestion can be supported by the recent reports that demonstrated the potential of SARS-CoV-2 infection to promote the reactivation of β-herpesviruses, such as CMV, which can cause opportunistic infection, including in patients with COVID-19, mainly in critically ill, due to some epiphenomena related to the severity of COVID-19, such as intensive care unit length-of-stay, inflammation, and/ or the treatment with corticosteroids [50-52]."

  1. Line 458-459  ‘monitoring of CMV IgG levels is important for patient follow-up by reflecting the infection status’

It will be helpful for reader to understand the point if authors can explain how CMV IgG levels reflect the infectious status in this report.

- Authors´ comments: We would like to thank you for the comment and inform you that we revised this part of the text and rewritten it as presented below.

"However, in a different way, the analysis of the correlation between the specific IgG levels for these viruses at the MO time-point showed a significant negative correlation, indicating that, in general, those older adults with better specific IgG response to SARS-CoV-2 presented lower levels of specific IgG for CMV. Despite it has been cited that CMV serostatus does not impact directly the response to vaccination magnitude [53], our finding corroborates some reports that point to the fact that CMV chronic infection status, which could be assessed by both humoral and cellular immune responses, can significantly interfere with the immune response to infection by other viral infectious agents [54], such as the Influenza virus [45] or even SARS-CoV-2, in the older adult population, particularly due to the immunosenescence and inflammaging [56]."

  1. Line 511, ‘By the way’ could be replaced to ‘Furthermore’ or ‘In addition,’.

- Authors´ comments: We would like to thank you for the comment and inform you that, as recommended, we altered "By the way" to "Furthermore".

  1. Line 516-517 'On the other hand, it was reported that the presence of allele 2 in IFN-lambda-3 gene (rs12979860) was related to more incidence of COVID-19 in Spain [65]. '

 Please speculate the reasons of these controversial data and discuss, which can suggest more direction of further study.

- Authors´ comments: We would like to thank you for the comment and inform you that we revised this part of the text and rewritten it as presented below.

" On the other hand, Saponi‑Cortes et al. [65] reported that allele 2 (TT), in the IFN-lambda gene (rs12979860), was significantly overexpressed in COVID-19 patients than in non-COVID-19 controls [65]. According to the same authors, and with some information previously cited, whereas the T allele has been associated with an ineffec-tive viral clearance, e.g., for HCV and other RNA viruses, the presence of the C allele could favor the clearance of these viruses [65, 28, 29]. Although it has been emphasized that some genetic polymorphisms, particularly in the IFN-lambda gene, could be involved in the susceptibility to COVID-19 [71], the discrepant results found until now in the literature reinforce the necessity of further studies to improve our under-standing of the impact of polymorphism in the IFN-lambda gene in the COVID-19 context."

Reviewer 2 Report

The relevance of genetics background from de host in the context of SARS-CoV-2 infection is an important issue that must be studied.

The information in this paper is of a very relevance in the understanding of the immune response against SARS-CoV-2 infection.

In the limitations of the study, the authors must include that they do not have the information about an active CMV infection determined by detection of the virus in a urine sample. And in conclusions, specially number 3, line 572, I really think the authors can not include that conclusion. They should performed a multivariate analysis in order to know the rol of CMV, and they can not perform that analysis because of the number of volunteers included in the study.

Include in the introduction and discussion some information about the polymorphism and HCV and the importance of CC favourable genotype in HCV clearence.

I recommend to the authors make that changes in writing in order to it can be published.

Author Response

Reviewer #2

Comments and Suggestions for Authors

The relevance of genetics background from de host in the context of SARS-CoV-2 infection is an important issue that must be studied.

The information in this paper is of a very relevance in the understanding of the immune response against SARS-CoV-2 infection.

- Authors´ comments: First of all, we sincerely would like to thank you for the criticism and also for your constructive comments/suggestions that enable us to improve this study. In addition, it is noteworthy to clarify that all alterations concerning your comments/suggestions in the main text are marked in blue.

In the limitations of the study, the authors must include that they do not have the information about an active CMV infection determined by detection of the virus in a urine sample. And in conclusions, specially number 3, line 572, I really think the authors can not include that conclusion. They should performed a multivariate analysis in order to know the rol of CMV, and they can not perform that analysis because of the number of volunteers included in the study.

- Authors´ comments: We would like to thank you for the comments and inform you that, as recommended, we revised the limitations of the study in order to include this suggestion related to the missing data on the presence or not of active CMV infection, as presented below.

"...4) the missing data regarding the presence or absence of active CMV infection in volunteers at different time points evaluated in this study..."

In relation to the comment about the conclusion, we would like to thank you for the suggestion and inform you that we revised and rewritten this part of the conclusion, as presented below.

"...and also that 3) the heterozygosis (C/T) and mainly the homozygosis for the allele 1 (C/C) in the il28b gene, which encodes an IFN-lambda-3 cytokine, could putatively be an important factor to be assessed in the specific antibody response to COVID-19 in the older adults seropositive for CMV."

Include in the introduction and discussion some information about the polymorphism and HCV and the importance of CC favourable genotype in HCV clearence.

- Authors´ comments: We would like to thank you for the comments and, as recommended, we added information regarding the IFN-lambda polymorphism and HCV, as presented below.

In the introduction section:

"Specifically, in terms of IFN-lambda, it has been reported that this cytokine, and also the single nucleotide polymorphisms (SNPs) linked to the cytokine IFNλ3 (also known as IL28B), can be involved not only in the spontaneous clearance of Hepatitis C virus (HCV) infection but also in the higher rates of sustained virological response associated with combined antiviral therapy [27, 28, 29]. In an interesting way, patients infected by HCV who carried the CC allele at the IFNλ3 polymorphic site presented a higher probability (3-times more) to eradicate this infection (both spontaneously and after antiviral therapy) than the individuals who showed the T genotype [28,29]."

In the discussion section:

"On the other hand, Saponi‑Cortes et al. [65] reported that allele 2 (TT), in the IFN-lambda gene (rs12979860), was significantly overexpressed in COVID-19 patients than in non-COVID-19 controls [65]. According to the same authors, and with some information previously cited, whereas the T allele has been associated with an ineffective viral clearance, e.g., for HCV and other RNA viruses, the presence of the C allele could favor the clearance of these viruses [65, 28, 29]."

I recommend to the authors make that changes in writing in order to it can be published.

- Authors´ comments: We would like to thank you for the comment and to inform you that, as recommended, a revision of the English (spelling and grammar) was carried out in order to improve it.

Reviewer 3 Report

In this manuscript, the authors analyzed the impact of IFN-lambda-3 polymorphism (rs12979860 C>T) on COVID-19-specific IgG responses in a cohort of 25 elderly residents of a long-care facility, selected for CMV IgG positivity. They demonstrated that both CMV seropositivity and the homozygosis for allele 1 (C/C) in the IFN-lambda-3 gene can negatively affect the antibody response to COVID-19 infection and vaccination in older adults.

Although the experiments were well performed, the manuscript is overall well-written and organized, and most of the conclusions are supported by the data presented, some improvements are required to optimize the publication.

The major and minor concerns are enlisted below: 

1) To strengthen the correlation between CMV/Sars-CoV-2 infection and IFN-lambda polymorphism, an important control is missing, i.e. CMV and Sars-CoV-2 negative volunteers. How is IFN-lambda polymorphism distributed in this population? I understand that at this point it would be difficult to recruit these kinds of subjects, given also the high seroprevalence of both viruses in the general population. However, it would be of interest at least to discuss this point.

2) Based on the results reported in Figure 1, panel C, The authors point out that Sars-CoV-2 infection may favor CMV reactivation. To better understand this aspect, it would be important to determine if MO sera are prior infections or reactivations, by CMV IgM serology or IgG avidity assays.

3) Abstract, lines 31 and 32: specify neutralizing antibodies against Sars-CoV-2.

4) Line 45: a comma should be replaced by a dot before ref 2.

Line 88-89: English revision is required.

5) Line 251 and Table 1: authors comment on the height difference between men and women. For consistency, also weight should be mentioned. Moreover, it would be of interest to include some more details about the cohort, such as ethnicity, diet habits, or other parameters that could influence the patients’ immunological response.

I hope my comments and suggestions are helpful.

Author Response

Reviewer #3

Comments and Suggestions for Authors

In this manuscript, the authors analyzed the impact of IFN-lambda-3 polymorphism (rs12979860 C>T) on COVID-19-specific IgG responses in a cohort of 25 elderly residents of a long-care facility, selected for CMV IgG positivity. They demonstrated that both CMV seropositivity and the homozygosis for allele 1 (C/C) in the IFN-lambda-3 gene can negatively affect the antibody response to COVID-19 infection and vaccination in older adults.

Although the experiments were well performed, the manuscript is overall well-written and organized, and most of the conclusions are supported by the data presented, some improvements are required to optimize the publication.

- Authors´ comments: First of all, we sincerely would like to thank you for the criticism and also for your constructive comments/suggestions that enable us to improve this study. In addition, it is noteworthy to clarify that all alterations concerning your comments/suggestions in the main text are marked in green.

The major and minor concerns are enlisted below: 

1) To strengthen the correlation between CMV/Sars-CoV-2 infection and IFN-lambda polymorphism, an important control is missing, i.e. CMV and Sars-CoV-2 negative volunteers. How is IFN-lambda polymorphism distributed in this population? I understand that at this point it would be difficult to recruit these kinds of subjects, given also the high seroprevalence of both viruses in the general population. However, it would be of interest at least to discuss this point.

-- Authors´ comments: We would like to thank you for the comments and clarify that we tried to obtain samples of individuals negative for CMV and Sars-CoV-2 during the time points assessed in this study, but, unfortunately, it was not possible.

In addition, we would like to inform you that, as suggested, we added information regarding the IFN-lambda polymorphism in CMV and SARS-CoV-2 in the discussion section, as presented below (page 14, lines 533, 534, 535, 536).

"On the other hand, Saponi‑Cortes et al. [70] reported that allele 2 (TT), in the IFN-lambda gene (rs12979860), was significantly overexpressed in COVID-19 patients than in non-COVID-19 controls [70]. According to the same authors, and with some information previously cited, whereas the T allele has been associated with an ineffec-tive viral clearance, e.g., for HCV and other RNA viruses, the presence of the C allele could favor the clearance of these viruses [70, 28, 29, 71]. Although it has been empha-sized that some genetic polymorphisms, particularly in the IFN-lambda gene, could be involved in the susceptibility to COVID-19 [71], the discrepant results found until now in the literature reinforce the need of further studies focused on to improve our understanding of the impact of polymorphism in the IFN-lambda gene in the COVID-19 context.."

" These data corroborate the literature since it has been reported that individuals with the genotypes CT or TT presented better control of CMV replication following primary infection than individuals presenting the genotype CC, due to the upregulation of in-terferon‐sensitive genes (ISGs), which acts in the inhibition of viral replication [72,73]"

2) Based on the results reported in Figure 1, panel C, The authors point out that Sars-CoV-2 infection may favor CMV reactivation. To better understand this aspect, it would be important to determine if MO sera are prior infections or reactivations, by CMV IgM serology or IgG avidity assays.

-- Authors´ comments: We would like to thank you for the comments and clarify that, as recommended, we carried out the CMV serology to evaluate the presence of specific IgM for this virus, and none of the volunteers presented this type of antibody. Therefore, we can suggest that all of them presented a chronic CMV infection. These pieces of information were added to the main text, as presented below.

In the "Material and Methods" section (page 05, lines 232, 233, 234, 235, 236):

"The seropositivity for CMV was evaluated through the assessment of the plasma level of specific IgM and IgG for CMV by using commercial ELISA test kits (BioClin, MG, Brazil). The CMV seropositivity of the volunteers has been defined as an IgM and IgG concentration ≥1.1 and ≥1.32 IU/ml, respectively, in agreement with the manufacturer’s instructions."

In the "Results" section (page 07, line 283):

"None volunteers presented specific IgM for CMV antigens."

In the "Discussion" section (page 12, lines 447, 448, 449, 450, 451, 452, 453, 454):

"Regarding the findings on specific IgM and IgG levels for CMV antigens, the observation that none of the volunteers presented detected IgM levels allows us to suggest that all of them were chronically infected by CMV. Besides, the significant elevation of specific IgG levels for CMV along with the time is very interesting and can allow us to putatively suggest that the SARS-CoV-2 infection could have altered the systemic inflammatory status favoring the inflammaging, then generating a favorable environment for CMV reactivation [19], which impacted the immune response and led to an increase of specific IgG levels for the CMV."

3) Abstract, lines 31 and 32: specify neutralizing antibodies against Sars-CoV-2.

-- Authors´ comments: We would like to thank you for the comments and inform you that, as suggested, we altered this part of the text.

4) Line 45: a comma should be replaced by a dot before ref 2.

Line 88-89: English revision is required.

-- Authors´ comments: We would like to thank you for the comments and inform you that, as recommended, we replaced the comma by a dot before the ref 2

In addition, we also would like to inform you that, as recommended, the text in the sentence presented in lines 88-89 was revised and rewritten, as presented below.

"Based on these data, it could be evidenced that, in this group of older adult individuals with a high number of CMV-reactive CD4+ T cells, the most prevalent respiratory virus was the coronavirus."

5) Line 251 and Table 1: authors comment on the height difference between men and women. For consistency, also weight should be mentioned. Moreover, it would be of interest to include some more details about the cohort, such as ethnicity, diet habits, or other parameters that could influence the patients’ immunological response.

- Authors´ comments: We would like to thank you for the comment and, firstly, inform you that we revised all the values presented in table 1, and the corrected values were added.

Besides, we also would like to inform you that we agreed that details about ethnicity could improve the present study, but, unfortunately, it was not possible to obtain these data. In relation to diet habits, we also agreed that this aspect can impact immunological response, but since all volunteers lived at the same institution during the study period, they were submitted to a similar diet provided by the institution. Based on these data, it could be evidenced that, in this group of older adult individuals with a high number of CMV-reactive CD4+ T cells, the most prevalent respiratory vi-rus was seasonal coronavirus.

I hope my comments and suggestions are helpful.

- Authors' comments: As previously mentioned, we are grateful for your excellent review and the comments and suggestions provided, which, certainly, allowed us to improve our study.

Round 2

Reviewer 3 Report

The Authors met my comments and modified the manuscript accordingly. The paper can now be accepted for publication.